# Analysis of the Model of Atherosclerosis Formation in Pig Hearts as a Result of Impaired Activity of DNA Repair Enzymes

**DOI:** 10.3390/ijms25042282

**Published:** 2024-02-14

**Authors:** Robert Paslawski, Paweł Kowalczyk, Urszula Paslawska, Jerzy Wiśniewski, Piotr Dzięgiel, Adrian Janiszewski, Liliana Kiczak, Maciej Zacharski, Barbara Gawdzik, Karol Kramkowski, Andrzej Szuba

**Affiliations:** 1Veterinary Insitute, Nicolaus Copernicus University in Toruń, Gagarina 7, 87-100 Toruń, Poland; urszula.paslawska@umk.edu.pl; 2WROVASC—Regional Specialist Hospital in Wroclaw, Research and Development Centre, Kamieńskiego 73a, 51-124 Wroclaw, Poland; piotr.dziegiel@umed.wroc.pl (P.D.); adrian.janiszewski@up.poznan.pl (A.J.); liliana.kiczak@upwr.edu.pl (L.K.); maciej.zacharski@upwr.edu.pl (M.Z.); szuba@yahoo.com (A.S.); 3The Kielanowski Institute of Animal Physiology and Nutrition, Polish Academy of Sciences, Instytucka 3, 05-110 Jabłonna, Poland; 4Department of Medical Biochemistry, Faculty of Medicine, Wroclaw Medical University, Chałubińskiego 10, 50-368 Wroclaw, Poland; jerzy.wisniewski@umed.wroc.pl; 5Department of Histology and Embryology, Wroclaw Medical University, Chałubińskiego 6a, 50-368 Wroclaw, Poland; 6Faculty of Veterinary Medicine, Life Science Institute, Poznań University of Life Sciences, Wojska Polskiego 28, 60-637 Poznań, Poland; 7Department of Biochemistry and Molecular Biology, Faculty of Veterinary Medicine, Wroclaw University of Environmental and Life Sciences, 31 Norwida St., 50-375 Wroclaw, Poland; 8Institute of Chemistry, Jan Kochanowski University, Świętokrzyska 15 G, 25-406 Kielce, Poland; b.gawdzik@ujk.edu.pl; 9Department of Physical Chemistry, Medical University of Bialystok, Kilińskiego 1, 15-089 Białystok, Poland; kkramk@wp.pl; 10Division of Angiology, Wroclaw Medical University, Pasteur 1, 50-367 Wroclaw, Poland

**Keywords:** oxidative stress, DNA damage, ethenoadducts, arteriosclerosis, endothelial dysfunction, heart

## Abstract

Excessive consumption of food rich in saturated fatty acids and carbohydrates can lead to metabolic disturbances and cardiovascular disease. Hyperlipidemia is a significant risk factor for acute cardiac events due to its association with oxidative stress. This leads to arterial wall remodeling, including an increase in the thickness of the intima media complex (IMT), and endothelial dysfunction leading to plaque formation. The decreased nitric oxide synthesis and accumulation of lipids in the wall result in a reduction in the vasodilating potential of the vessel. This study aimed to establish a clear relationship between markers of endothelial dysfunction and the activity of repair enzymes in cardiac tissue from a pig model of early atherosclerosis. The study was conducted on 28 female Polish Landrace pigs, weighing 40 kg (approximately 3.5 months old), which were divided into three groups. The control group (*n* = 11) was fed a standard, commercial, balanced diet (BDG) for 12 months. The second group (*n* = 9) was fed an unbalanced, high-calorie Western-type diet (UDG). The third group (*n* = 8) was fed a Western-type diet for nine months and then switched to a standard, balanced diet (regression group, RG). Control examinations, including blood and urine sampling, were conducted every three months under identical conditions with food restriction for 12 h and water restriction for four hours before general anesthesia. The study analyzed markers of oxidative stress formed during lipid peroxidation processes, including etheno DNA adducts, ADMA, and NEFA. These markers play a crucial role in reactive oxygen species analysis in ischemia–reperfusion and atherosclerosis in mammalian tissue. Essential genes involved in oxidative-stress-induced DNA demethylation like OGG1 (8-oxoguanine DNA glycosylase), MPG (N-Methylpurine DNA Glycosylase), TDG (Thymine-DNA glycosylase), APEX (apurinic/apirymidinic endodeoxyribonuclease 1), PTGS2 (prostaglandin-endoperoxide synthase 2), and ALOX (Arachidonate Lipoxygenase) were measured using the Real-Time RT-PCR method. The data suggest that high oxidative stress, as indicated by TBARS levels, is associated with high levels of DNA repair enzymes and depends on the expression of genes involved in the repair pathway. In all analyzed groups of heart tissue homogenates, the highest enzyme activity and gene expression values were observed for the OGG1 protein recognizing the modified 8oxoG. Conclusion: With the long-term use of an unbalanced diet, the levels of all DNA repair genes are increased, especially (significantly) Apex, Alox, and Ptgs, which strongly supports the hypothesis that an unbalanced diet induces oxidative stress that deregulates DNA repair mechanisms and may contribute to genome instability and tissue damage.

## 1. Introduction

Hyperlipidemia causes oxidative stress, an additional factor increasing the risk of acute cardiac events [1,2,3,4,5,6,7,8,9,10]. Oxidative stress develops in situations of excessive production of reactive oxygen species (ROS) and both the accompanying free radicals as well as non-radical organic compounds containing reactive functional groups (i.e., amines, ketones, hydroxyl compounds, and aldehydes) [11,12,13,14,15,16,17,18,19,20,21,22]. Under physiological conditions, the harmful effects of ROS are prevented through many mechanisms [23,24,25,26,27,28,29,30]. Increasing experimental and clinical evidence indicates that uric acid plays an important in vivo role as an antioxidant in cardiovascular disease [31,32,33,34,35,36,37,38,39,40,41]. However, the precise role of uric acid in oxidative stress damage is not entirely clear [42,43,44,45,46,47,48]. ROS damage cellular structures and macromolecules and participate in the formation of DNA adducts, which inhibit the replication process, cause DNA strand breaks, and induce apoptosis [12,13,14,15,16,17,18,19,20,21,22,23,24,25,26,27,28]. The highest number of modifications occur in the case of guanine, adenine, and cytosine, and the least in the case of thymine, because both of its forms, lactam and lactim, are very weakly reactive. One of the most known guanine adducts is 8-oxo guanine (8-oxoG), characterized by a high miscoding potential [28,29,30,31,32,33,34]. All these DNA damages may weaken the cell’s repair potential, inhibit its ability to properly transcribe the damaged DNA fragment, or contribute to the persistence of potentially dangerous mutations in the cell’s genome [35,36,37,38,39]. Multiple DNA repair systems prevent the accumulation of unwanted mutations [40,41,42,43,44,45,46,47,48,49,50,51].

Scientific research also indicates that diet may induce or weaken DNA repair mechanisms including the process of atherosclerosis [52,53,54,55,56,57,58,59,60,61,62,63,64,65]. Therefore, we focused on examining the association of selected markers of endothelial dysfunction, lipid peroxidation, and DNA repair enzyme activity in porcine heart tissue with early atherosclerosis induced by long-term unbalanced nutrition [66,67,68,69,70,71,72,73,74]. The study’s second aim was to analyze the impact of switching to a balanced diet [75,76,77,78,79,80,81,82,83,84,85,86].

## 2. Results

All pigs in the final examination had higher end-systolic and end-diastolic blood pressure than at the beginning of the experiment, but in none of the groups, this difference reached the level of statistical significance (even in the RG, where it was the largest). There was also no difference in the thickness of the heart wall between the groups during the experiment (Table 1), while in the UDG compared to the other groups, an increase in IMT in the femoral artery was observed. In the last study, it was significantly higher compared to the RG (Appendix A).

In all groups of pigs, CRP levels, white blood cell counts, and proinflammatory cytokine levels showed no difference between the groups. The highest levels of etC, etA, and 8oxoG in the heart muscle tissue, blood glucose, serum insulin, NEFA, and the Homa index were observed in the RG (Figure 1 and Table 1).

Serum uric acid concentrations were higher in both groups of pigs fed the unbalanced diet (UDG and RG) as compared to the BDG, but the difference only reached statistical significance after twelve months of the experiment in the RG (Table 1).

We found several correlations of uric acid concentration with other parameters in the UDG. It negatively correlated with the maximal (rs= −0.65) and minimal (rs= −0.59) diameter of the femoral artery in the UDG (Figure 2) and positively with etC (rs = 0.5) and TBARS (rs = 0.62), which are oxidative stress markers in the heart (Figure 3).

In all groups of pigs, the concentrations of L-arginine at the beginning and the end of the study were similar. However, the concentration of Asymmetric Dimethylarginine (ADMA) and Symmetric Dimethylarginine (SDMA) at the beginning were about two times higher than at the end of the study. There was a significant increase in ADMA in the RG vs. the BDG and an increase in SDMA in the UDG vs. the BDG (Table 1). The levels of ADMA and SDMA correlated positively in the UDG with TGC (for both r_s_ = 0.79). In the UDG, ADMA concentration positively correlated with HDL (r_s_ = 0.91), and SDMA positively correlated with LDL (r_s_ = 0.86).

In none of the groups of pigs, based on urine examination and abdominal ultrasound, did we detect any abnormalities in the structure and function of the kidneys or urinary tract. The concentrations of urea and creatinine in serum and urine remained within physiological limits (Table 1). From the beginning of the experiment, feeding with a high-fat diet resulted in persistently low leptin concentrations (Figure 4).

As previously described [41], the thickening of the IMT complex and the presence of grade I and II atherosclerotic plaques were demonstrated in the femoral artery of pigs fed with UD. The same changes were observed in the coronary arteries of these pigs.

The statistical analysis showed a correlation between VEGF levels (VEGF-A, VEGF-B, VEGF-C) with the IMT of the femoral artery in the UDG (Figure 5).

### 2.1. mRNA Level of Genes Encoding DNA Repair Enzymes

#### 2.1.1. mRNA Abundance of DNA Repair Enzymes

We found that genes coding all investigated enzymes of the DNA repair system (MPG, TDG, OGG1, ALOX, APE1, PTGS2) were expressed in all analyzed tissues (Figure 6). Differences in the number of analyzed transcripts between the analyzed groups are presented in Figure 6. The observed levels of gene expression were the highest for the OGG1 gene and comparable with the mRNA values for the APEX, ALOX, and PTGS2 genes (Figure 6). All results were statistically significant at the level of *p* < 0.05–*p* < 0.01.

#### 2.1.2. Analysis of DNA Repair after Fpg Cleavage

In all analyzed groups of heart tissue homogenates, the highest activity values after Fpg cleavage were observed for the OGG1 protein recognizing the modified 8oxoG as compared to the other analyzed DNA damages (modification) recognized by TDG and Mpg glycosylases (Figure 7). The highest values of damage were found, exceeding 3.7% and thus corresponding to high guanine oxidations.

## 3. Discussion

Our animal model of the natural, slow development of atherosclerosis provides information that could not be studied in humans in the early stages of atherosclerosis—particularly about molecular changes in heart and artery tissue. In addition, it enables omitting factors that make result interpretation difficult, such as the varied composition of a high-calorie diet (differences in the amount of fat and the proportions of individual fatty acids and the number of carbohydrates), different physical activity and stress levels, genetic factors, the presence of other diseases (including diabetes and hypertension), an unknown moment of starting an unhealthy diet or the onset of metabolic disorders, and the influence of drugs used. The choice of the pig as an animal model is justified by the natural predisposition of pigs to excessive consumption, omnivorousness, the most similar lipid metabolism, and the greatest similarity to human anatomy and physiology of the circulatory system [40]. As previously described, pigs fed an unbalanced diet developed metabolic disorders, hyperinsulinemia, and dyslipidemia (hypercholesterolemia, hypertriglyceridemia), but this was not a typical metabolic syndrome due to the lack of fasting hyperglycemia [10]. These disorders were enough to cause the pathological remodeling of the arterial wall, including the hypertrophy of the intima and middle layer of the artery, reduction in the artery diameter, and the presence of I^0^ and II^0^ atherosclerotic plaque in the peripheral artery [41,80]. Moreover, analyzing previous studies, this incomplete metabolic syndrome seems to be typical of the porcine model [40].

We think that this may be due to the properties of pig feed, which does not cause sudden increases in glucose in the blood serum. Therefore, we included leptin in the study panel. The task of leptin is to inform the nerve centers that regulate appetite and energy expenditure about the nutritional status and amount of fat tissue [81]. It has been shown that the concentration of leptin is higher in obese people and is proportional to the degree of obesity [82]. The finding of a positive correlation between insulin resistance and leptin concentration prompted the use of its concentration as a marker in the diagnosis of obesity and diseases related to it [83]. It has been shown that leptin concentration decreases with long-term hyperglycemia and, therefore, long-lasting hyperglycemia promotes the maintenance of appetite by lowering leptin concentration [84]. Our research confirmed that in improperly fed pigs, although the leptin level progressively increased quite significantly, the difference was not large enough to achieve statistical significance.

We have previously reported that pigs fed an atherosclerotic diet (UDG) showed significant modification of intima–media gene expression and an increase in glucose concentration in a femoral artery wall sample despite physiological blood glucose levels [82].

Atherosclerosis is considered to be a progressive systemic inflammatory disease, mainly affecting the walls of the aorta, carotid, and coronary arteries, with a long latency period [57,58]. Such a slow inflammatory process cannot be recognized in standard tests: our animals had normal body temperature, leukocyte counts, and CRP levels. No inflammatory infiltrates were found in the walls of peripheral and coronary arteries. We also did not demonstrate a significant increase in the concentration of proinflammatory cytokines in blood serum. Other authors also reported no increase in the concentration of proinflammatory cytokines IL-1, IL-6, and TNFα in the walls of atherosclerotic arteries [39]. Another surprising observation was the fact that the highest mean NEFA values (a marker of prediabetes) were observed in the RG and not, as expected, in the UDG. It is therefore possible that the very fact of a sudden change in diet, even if it is a change to a healthier diet, causes metabolic disorders.

After documenting the effectiveness of the applied model in inducing atherosclerotic changes, the impact of reactive oxygen species on myocardial lipid peroxidation products (TBARS) and the amount of nucleic acid repair enzymes was assessed. It seems that at such an early stage of disorders, lipid peroxidation is a mechanism that protects the genetic material from damage. A similar phenomenon has already been described—research shows that LDL increases the expression of the protein methyltransferase 1, which is responsible for L-arginine methylation and the synthesis of ADMA [57,60]. In the pigs studied, the relationship between ADMA and cholesterol concentration was visible in the improperly fed pigs (UDG), while in the BDG and RG, no such relationship was observed. ADMA is an endogenous, competent inhibitor of all isoforms of nitric oxide synthase (NOS), including endothelial NOS (e-NOS). ADMA plays a role in controlling vascular relaxation. The suppression of e-NOS activity reduces the atherosclerotic properties of the endothelium. Since ADMA is formed as a result of the degradation (hydrolysis) of arginine-rich proteins, its level in the blood serum may be significantly influenced by the supply of arginine in the diet and, consequently, the concentration of L-arginine in the blood serum [60]. In the pigs studied, the baseline L-arginine concentration was similar in all groups; therefore, the differences in serum ADMA and SDMA concentrations observed at the end of the study were due to factors other than the amount of L-arginine, i.e., disorders caused by poor nutrition. It is supposed that ADMA impairs endothelial function not only by inhibiting eNOS activity but also by increasing the synthesis of peroxides [45,46,47,48,49,50,51,52]. The importance of the relationship between ADMA and peroxides has been documented in large clinical trials that have confirmed that antioxidant therapy reduces serum ADMA levels [45]. This relationship has been confirmed in experimental studies in which the incubation of human umbilical vein endothelial cells with oxidized low-density lipoprotein increases the ADMA content in the culture medium [77]. ADMA appears to impair endothelial function regardless of the coexistence of insulin resistance or left ventricular hypertrophy [47]. Therefore, the relationship between ADMA and IMT has been observed in adults without overt cardiovascular disease [47] and was found to be stronger than the relationship with LDL [57]. Our research results confirm these previous observations.

In all groups of pigs, the highest concentration of ADMA was observed at the beginning of the study, and then it decreased. ADMA is considered to be one of the better markers of risk stratification in patients with cardiovascular disease, regardless of age, gender, or weight. This is due to its (generally) constant production resulting from proper protein metabolism in many tissues [45,46,47,48,49,50,51,52,60,61,62]. Our results indicate that ADMA levels may be elevated when metabolism is intensified, such as in the period of rapid growth in pigs. An indirect confirmation of our hypothesis seems to be the results of the studies by Hermenegildo et al., who showed an increased level of ADMA in patients with hyperthyroidism [61]. A comparison of the results of the final examination showed the highest concentration of ADMA in the RG, while we expected this in the UDG. This observation is consistent with our previous metabolomics studies, conducted on the same pigs, in which, also, the greatest metabolic differences compared to the control group were found in the RG [41]. SDMA is a structural isomer of ADMA and does not inhibit NOS [52]. We observed the highest level of SDMA in the UDG. Its known that SDMA is competing with l-arginine for transport across cell membranes [63]. Several studies have shown that SDMA itself does not play any physiological role [52]. However, more and more evidence supports the involvement of SDMA in the development of inflammation and atherosclerosis [64]. SDMA is supposed to be responsible for increasing the production of reactive oxygen species (ROS) by monocytes [57]. ADMA also indirectly enhances ROS production: high ADMA levels result in reduced NO availability, which increases the expression of vascular cell adhesion molecule-1 (VCAM-1) (mainly due to the increased synthesis of the transcription factor NF-κB). Other effects of this dysregulation are increased ROS production and an increase in CRP concentration [51]. As mentioned above, there was no increase in CRP or other signs of inflammation, so we believe that the latter mechanism—even if it occurs—does not play a significant role. VCAM-1 adhesion molecules, together with MCP-1, ICAM-1, E-selectin, and P-selectin, attracted circulating monocytes toward the atherosclerotic lesion. VCAM-1 prompts the maturation of monocytes into proinflammatory macrophages (M1 phenotype) [65]. In normal conditions, macrophages regulate lipoprotein metabolism by binding oxidized LDL and prompting the uptake of these proteins into the cell. Macrophages in the vessel wall are loaded with oxidized LDL particles, leading to the formation of foam cells [51]. The endothelial atherogenic phenotype has an increased permeability to circulating LDL, and their accumulation in the tunica intima is the first step in plaque formation. LDL is exposed to oxidation, producing oxidized LDL (oxLDL), acting as damage-associated molecular patterns (DAMPs), damaging the endothelium, and triggering the inflammatory process by binding to pattern recognition receptors (PPRs) [59]. As we mentioned above, under physiological conditions, the harmful effects of ROS are prevented by many mechanisms that limit the harmful effects of ROS. Antioxidant properties are also attributed to uric acid. In our study, pigs with higher levels of oxidative stress (UDG and RG) had higher uric acid concentrations (Table 1), and uric acid positively correlated with etC and TBARS—markers of oxidative stress in the heart (Figure 3). In this light, an increase in uric acid levels seems to be beneficial as an expression of well-functioning compensatory mechanisms.

VEGF is a family of heparin-binding proteins involved in reducing oxidative stress, inhibiting inflammation, and the progression of atherosclerosis by promoting the regulation of lipid metabolism, dilation, and proliferation of lymphatic vessels [85]. The VEGF family consists of five human gene products. Three of them—VEGF-A, VEGF-B, and placental growth factor (PIGF)—regulate the growth of blood vessels, and two—VEGF-C and VEGF-D—modulate lymphangiogenesis, thus participating in the regulation of lipid metabolism through the lymphatic system of the body (via unidirectional absorption and the transport of lipids from the digestive tract to the venous system) [4]. Moreover, VEGF-C is involved in the repair of damaged heart muscle [86]. Based on our studies, it appears that although the increase in blood pressure, heart wall hypertrophy, and oxidative stress seemed to be insignificant, the sum of these small changes already resulted in a significant increase in VEGF-C concentration. We also consider it an expression of well-functioning protective mechanisms at this stage of the disease.

Our study documented the activity of the DNA repair enzymes responsible for the excision of damaged nucleobases in the heart. We observed an increased level of repair enzymes towards etC in the RG. The results confirmed that high oxidative stress, as shown by TBARS concentration, was associated with a high level of DNA repair enzymes. Tissue repair capacity largely depends on the expression of genes participating in the repair pathway. We investigated the mRNA levels of DNA glycosylases excising etA (MPG), etC (TDG), 8-oxoG (OGG1), ALOX, APE1, and PTGS2, as well as that of the next enzyme in the BER pathway, Ape1. The latter can activate 8-oxoG and etC excision in vitro up to 12-fold by increasing the enzyme turnover on damaged DNA [19,20]. This suggests that the pathology of endothelial dysfunction in the swine model of early arteriosclerosis may be induced by oxidative stress. The highest levels of expression were observed for the Apex and Alox genes, followed by the 8-oxoG (OGG1) gene, which encodes bi-functional glycosylases. These enzymes compete with each other for the site of damage through the active center in the helix–herpin–helix domain, attaching to the modified bases and changing their spatial rearrangement [42]. OGG1, a housekeeping gene [42], was shown to be induced by oxidative stress [34,35,36,37,38,39]. The lipid peroxidation mobilizes the entry of existing OGG1 molecules from the cytoplasm to the nucleus and later may also stimulate its synthesis de novo. The 8-oxoguanine DNA glycosylase (OGG1) is important because it removes from double-stranded DNA to a broad spectrum of oxidized and alkylated bases: 7,8-dihydro-8-oxoguanine (8-oxoguanine), 8-oxoadenine, and unsubstituted and substituted imidazole ring-opened purines introduced into DNA, both by hydroxyl radicals as well as by chemical carcinogens including anticancer drugs [34,35,36,37,38,39]. The Fpg protein has two additional activities: (i) the AP-lyase activity which cleaves both 3′ and 5′ to the AP site, thereby removing the AP site and leaving a one-base gap through alpha–beta-elimination, and (ii) a dRPase activity, which removes the 5′ terminal deoxyribose phosphate from DNA incised by an AP endonuclease [34,35,36,37,38,39].

MPG is regulated in the cell cycle, being induced at the end of G1 and in the S phase [34,35,36,37,38,39] and by different exogenous factors like ionizing radiation and other ROS-generating factors [35]. In our study, the transcription of TDG was also induced moderately. The biggest induction was observed for AP-endonuclease, Ape1. Ape1 was shown to stimulate the excision of 8-oxoG and etC by increasing OGG1 and TDG turnover on damaged DNA [34,35,36,37,38,39]. This can explain the increased repair capacity for 8-oxoG and etC in pigs after dietary treatment, even though glycosylases’ mRNA level did not differ significantly from that in control animals. etA glycosylase, being monofunctional, requires Ape1 activity and, therefore, in the assay used, also depends on its availability. The transcription of Ape1 is known to be induced by oxidative stress [34,35,36,37,38,39], and has also been shown to increase in many types of human cancers [34,35,36,37,38,39]. The acceleration of DNA repair, similarly to its inhibition, may, nonetheless, be a deleterious phenomenon. It was shown that in tissues from noncancerous colons of ulcerative colitis patients, MPG and APE1 were significantly increased and microsatellite instability (MSI) was positively correlated with the enzymes’ imbalanced repair activities. The latter results were supported by mechanistic studies using yeast and human cell models, in which the overexpression of MPG and/or APE1 was associated with frameshift mutations and MSI [21,69,70,71,72,73,74,75,76,77,78,79,80,81,82,83]. These results are consistent with the hypothesis that the adaptive and imbalanced increase in MPG and APE1 is a novel mechanism contributing to MSI, and, more generally, to genome instability, which, in consequence, may extend to other diseases with MSI, e.g., to cancer. Our results seem to confirm this hypothesis, suggesting that imbalanced DNA repair contributes to increased genomic instability and the formation of lesions in tissues [69,70,71,72,73,74,75,76,77,78,79,80,81,82,83,84,85,86].

Understanding the genotoxic properties of endogenous DNA damage as etheno DNA adducts, such as 1,*N*^6^-ethenoadenine (εA), 3,*N*^4^-ethenocytosine (εC), *N*^2^,3-ethenoguanine (εG), and 1,*N*^2^-ethenoguanine (εG) and repair pathways, resulting in the so-called oxidative stress process, is fundamental to understanding the mechanisms of diseases that depend on chronic inflammation. Due to the large range of these products and pleiotropic action, knowledge about the molecular mechanisms of their action is still fragmentary [34,35,36,37,38,39].

Among the most genotoxic oxidative DNA damages are 1, *N^6^*-ethenoadenine (εA), 3, *N^4^*-ethenocytosine (εC), and 8-oxoguanine (8-oxoG). All these lesions have a high miscoding potential and are found in the DNA of untreated animals and humans. The changes in the heart caused by a pro-atherogenic diet may induce different levels of ethenoadducts in DNA, which may also increase Wilson disease and primary hemochromatosis, diseases leading to the development of ischemic heart events. For example, the 8-oxoG level was found to be increased in the blood leukocytes of lung cancer patients in comparison with healthy controls due to decreased repair capacity. Decreased repair of εA and εC was also shown to accompany the development of a specific histological type of heart cancer, the etiology of which may lead to chronic inflammations induced DNA modifications as both 8-oxoG as well as εA and εC. These damages are eliminated from DNA by the Base Excision Repair (BER) pathway, initiated by DNA glycosylases, which cleave the N-glycosidic bond between the damaged base and the deoxyribose moiety, leaving behind a baseless sugar (AP-site). AP-sites are then processed by AP-endonuclease. DNA glycosylases excising 8-oxoG (OGG1) and εC (TDG) have a high affinity for reaction products and do not leave the AP-site without the assistance of AP-endonuclease [19,20].

Based on our research results related to DNA repair in the nicking assay method and gene expression analysis, it was noticed from literature data that ethenoadenine, which is excised from DNA by N-methylpurine-DNA glycosylase (MPG), which also repairs a broad spectrum of DNA alkylation products like 3-methyladenine or 7-methylguanine, hypoxanthine, ethylated bases, and 1,*N^2^*-etG, was present. Thymine-DNA glycosylase (TDG) excises εC and thymine from G:T pairs, resulting from deamination of 5-methylcytosine. OGG1 glycosylase, in addition to 8-oxoG, removes also unsubstituted and substituted imidazole ring-opened purines. OGG1 is endowed with AP-lyase activity and cleaves AP-sites via β-elimination. The major human 5′AP-endonuclease is APE1 (also called HAP-1/APX/REF1), which is strongly bound with the activity and expression of the OGG1 gene. In addition to its AP-endonuclease activity, APE1 also acts as a redox regulation enzyme for a transcription factor, as well as exhibits other activities, like 3′-5′exonuclease, phosphodiesterase, 3′phosphatase, and RNase H activities [34,35,36,37,38,39,69,70,71,72,73,74,75,76,77,78,79,80,81,82,83,84,85,86].

Thus, oxidative stress including diet after its administration mobilizes the entry of existing OGG1 molecules from the cytoplasm to the nucleus and later may also stimulate its synthesis de novo. MPG is regulated both in the cell cycle, being induced at the end of G1 and in the S phase, and by different exogenous factors, like ionizing radiation and other ROS-generating factors. In our study, the transcription of TDG was also induced moderately. The biggest induction after OGG1 in the expression level was observed for AP-endonuclease (Ape1). Ape1 was shown to stimulate the excision of 8-oxoG and εC by increasing OGG1 and Tdg turnover on damaged DNA. This can explain the increased repair capacity for 8-oxoG and εC of in the cardiac muscle of pigs in groups fed a balanced diet (BDG) and unbalanced diet (UDG) and in pigs fed 9 months of an unbalanced diet followed by 3 months of a balanced diet (RG) [44,53].

Even from 3 months to 9 months after different types of diet induction, glycosylases’ mRNA level did not differ significantly from that in control animals. εA glycosylase, being monofunctional, requires Ape1 activity, and, therefore, the assay used also depends on its availability. The transcription of Ape1 is known to be induced by oxidative stress and was also shown to increase in many types of human cancers [34,35,36,37,38,39]. This suggests that the treatment of pigs with a balanced diet (RG) induces long-term oxidative stress, which not only may increase DNA damage but also induces repair processes. The acceleration of DNA repair, similarly to its inhibition, may, nonetheless, be a deleterious phenomenon. These results are consistent with the hypothesis that the adaptive and imbalanced increase in MPG and APE1 is a novel mechanism contributing to MSI, and, more generally, to genome instability, and, in consequence, may extend to other diseases with MSI, e.g., to cancer.

## 4. Materials and Methods

### 4.1. Ethics Statement

The study was approved by the second Local Ethics Committee in Wroclaw (resolution no 23/2009 date 23 September 2009).

### 4.2. Animals and Diets

As described previously [41], 28 female pigs (Polish Landrace) of 40 kg body weight (about 3.5 months of age) were divided into three groups. Pigs in the control group (*n* = 11) were fed, for 12 months a standard, commercial, balanced diet (BDG) containing the following: 2100 kcal/kg, 14.7% crude protein, 3.1% fat, 4.7% crude fiber, 90.44% dry mass, 6.06% ash, 0.5% NaCl, 1.05% Ca, 0.77% P, 0.62% lysine, 0.24% methionine, 0.3% cystine, 0.48% threonine, 0.183% tryptophan, 13,243 IU/kg vit. A, 2000 IU/kg Vit. D_3_, 81.65 mg/kg Vit. E, 4.11 mg/kg Vit. B_1_, 7.16 vit. B_2_, 50.22 mg/kg Vit. PP, 24.29 mg/kg Vit. B_5_, 6.11 mg/kg Vit. B_6_, and 36 µg/kg Vit. B_12_. This feed had a low sugar content. The control group had limited access to food, which was proportional to body mass, with a maximum calorie intake of 4200 kcal/pig/day.

In the second group (*n* = 9), animals were given a Western-type high-calorie, unbalanced diet (UDG) containing 2588.12 kcal/kg, 13% crude protein, and 16% fat. High energy concentration was obtained through the inclusion of beef tallow (14%) and saccharose (4%) in the diet. Pigs in this group had unlimited access to feed and consumed approximately 4 kg/pig/day during the period of highest food intake [41].

The third group of pigs (*n* = 8) was fed a Western-type diet for nine months and then was given a standard, balanced diet (regression group, RG). All pigs had unlimited access to water throughout the whole experimental period. Animals were housed in a single room divided into three pens (one group/pen) with a temperature of 18–20 °C and 60–75% humidity.

### 4.3. Clinical Evaluation

Every three months, a control examination was conducted under identical conditions with food restriction for 12 h and water restriction for four hours before the general anesthesia. After 12 h of fasting, pigs were premedicated with an intramuscular injection of a mixture containing the following: medetomidine hydrochloride, 1 mg/m^2^ body surface (Cepetor, CP-Pharma, Burgdorf, Germany), 5 mg/m^2^ BSA midazolam (Midanium, WZF Polfa, Warsaw, Poland), and 264 mg/m^2^ BSA of ketamine (Bioketan, Vetoquinol Biowet, Gorzów Wielkopolski, Poland). Each pig was intubated and placed in the left lateral position with the front legs slightly stretched forward in a room whose temperature was set at 21 °C. The vital signs of each pig, namely tongue pulse oximetry, non-invasive blood pressure, respiration rate, and body temperature, were monitored using a LIFEPAK^®^ 12 multiparameter monitor (Physio-Control, Inc., Redmond, WA, USA).

### 4.4. Measurements of Arterial Blood Pressure

Measurements of arterial blood pressure were performed on the common digital artery after a 20 min rest period using model 811-B of the Doppler Flow Detector (Parks Medical Electronics Inc., Aloha, OR, USA). An average of 3–5 consecutive SAP measurements was recorded.

### 4.5. Blood Examination

The complete blood count (CBC) was analyzed using an ABC-Vet analyzer while the biochemical analyses (glucose, alanine and aspartate aminotransferases, urea, creatinine, and total protein) were performed using the MAXMAT PL Biomedical Analyzer to exclude inflammatory, liver, and kidney disease.

SPINREACT enzyme tests were used to determine total cholesterol and triglycerides in blood serum. For cholesterol measurements, blood was tested twice—at the beginning and the end of the study. QUANTOLIP™ LDL Calibrator Set tests by Immuno AG (Vienna, Austria) were used to evaluate the high-density lipoprotein (HDL) subclass. HDL cholesterol concentrations were performed using SPINREACT enzyme tests. Readings of the concentrations of the determined parameters were made on a Beckman DU-650 spectrophotometer (Brea, CA, USA). LDL values were calculated using Friedewald’s formula.

### 4.6. Measurements of Insulin, Leptin, Vascular Endothelial Growth Factor (VEGF), C-Reactive Protein (CRP), Uric Acid, and Glucose

The enzyme-linked immunosorbent assays (ELISA) and the colorimetric methods measured using the Synergy LX apparatus (Biokom, Lodz, Poland) were used to quantify the concentrations of the tested factors. All determinations were performed using commercially available reagent kits according to the instructions provided by the manufacturers. The following tests were used for the determination: Pig Leptin ELISA Kit by EIAab (Wuhan, China), VEGF-A ELISA Kit by EIAab (catalog no. E0143p), VEGF-C ELISA Kit by EIAab (catalog no.—D ELISA kit by EIAab (Cat # E0146h), Pig CRP Elisa from Immunology Consultants Laboratory (Portland, OR, USA, Cat # E-5CRP), Mercodia Pig Insulin ELISA (Uppsala, Sweden, Cat # 10-1200-01), Uric Acid Test Kit from Bio Vision (Milpitas, CA, USA, Cat # K608-100), Cayman Chemical Company Creatine Kit (Ann Arbor, MI, USA, Cat. No. 700460), and Cayman Chemical Company Colorimetric Glucose Kit (Cat. No. 1000958). The analyzed samples were calculated with MARS Data Analysis Software V3.01 R2 with reference to the obtained standard curves. Homeostasis model assessment of insulin resistance (HOMA) was calculated using the following formula: fasting insulin concentration (mU/mL) × fasting glucose (mmol/L).

### 4.7. Cytokines Measurement

Fresh-frozen serum samples (−80 °C) collected directly before euthanasia were used for interleukin-1β (IL-1β), interleukin-6 (IL-6), and tumor necrosis factor-α (TNF-α) measurements with Quantikine ELISA kit (R&D Systems, Minneapolis, MN USA) in accordance with the manufacturers’ instructions. The results were expressed in pg/mL units.

### 4.8. Urine Examination

The urine examination was carried out immediately after receiving the urine samples (via cystocentesis) to exclude kidney and urinary tract disease. A physicochemical examination of urine included an analysis of its color, transparency, specific gravity, pH, protein, albumin, creatinine, glucose, blood, acetone, and urobilinogen levels. The urine sediment was obtained through centrifugation for 10 min at 2500× *g*. The urine sediment was detected using a ZEISS Primotech microscope (Oberkochen, Gremany), measuring the numbers of epithelial cells, blood cells, casts, and bacteria.

### 4.9. Electrocardiographic Examination

The electrocardiogram (ECG) in each pig was recorded when the pig was in the left lateral position using a BTL SD08 electrocardiograph (Prague, Czech Republic) with noise and muscle tremor filtering.

### 4.10. Echocardiographic Measurements

Transthoracic echocardiography was performed according to the procedure described earlier [66], i.e., in the position lying on the left side, using the Aloka Alfa 7 (Tokyo, Japan) ultrasound machine, with a sector transducer with a frequency of 3.5–5 MHz. The transducer was placed just above the elbow tubercle in the fourth intercostal space. The examination was performed using one- and two-dimensional imaging, obtaining standard cross-sections from the right and left sides of the chest. ECG was recorded simultaneously. End-diastolic measurements were made on the level of the top of the R wave and end-systolic measurements were made at the end of the T wave. End-diastolic thickness (IVSd) of the interventricular septum, end-diastolic (LVIDd) and end-systolic (LVIDs) internal diameter of the left ventricle, and the end-diastolic thickness of the free wall of the left ventricle (LWd) were assessed from the right parasternal four chambers. Based on these measurements, shortening fraction (FS) and ejection fraction (EF) were calculated.

Each ultrasound measurement was performed three times and the average values were taken for further analysis, and these values were compared with the physiological parameters of pigs [66].

### 4.11. Ultrasonographic Examination of the Kidney

A standard abdominal ultrasound kidney examination was carried out using the Hitachi Aloka F37 Japan machine with a 5–10 MHz linear probe to exclude kidney diseases.

### 4.12. The Tissue Collection

After 12 months of the experiment, animals were presented for euthanasia. Tissue sections from the left ventricle (LV) free wall were taken and immediately frozen in liquid nitrogen. Samples were stored at −80 °C before further analyses.

### 4.13. The LC-MS/MS Analysis of DMA, ADMA and SDMA

Samples, weighing 100 mg, of porcine myocardium were transferred into 2 mL polypropylene tubes and mechanically homogenized for 1 min in 200 µL of water. Then, 10 µL of internal standard solutions (IS D6-DMA Urinary Dimethylamine (50 µM); D7-ADMA Asymmetric Dimethylarginine (20 µM); D7-Arg (100 µM)) and 500 µL of acetonitrile were added and sonicated in ice-cold bath for 1 min using the following settings: 50% power output, 5 s ON/5 s OFF pulses. Samples were vortexed (5 min, 25 °C) and centrifuged (5 min, 4 °C, 21,000 RCF). Then, 300 µL of supernatant was transferred into 2 mL polypropylene tubes; 10 µL of sodium carbonate (20 mM) and 10 µL of 10% PFBoylCl in acetonitrile were added. Samples were vortexed (5 min, 25 °C) and centrifuged (7 min, 4 °C, 21,000 RCF), and 100 µL of supernatant aliquots was transferred into 2 mL glass vials and diluted two-fold with water.

The LC-MS/MS analysis was performed on a 1260 Infinity LC system combined with a 6420 Triple quadrupole mass spectrometer (Agilent Technologies, Santa Clara, CA, USA) [67]. Both the level of SDMA (Symmetric Dimethylarginine) and ADMA depend on the kidney function; therefore, to exclude kidney disease, an ultrasound examination of the kidneys and examination and determination of the concentrations of urea and creatinine in the blood serum were performed.

#### Analysis of Cholesterol and Triacylglycerol Concentrations in Cardiac Muscle

The analysis was performed as described previously but using the frozen cardiac muscle tissue [68]. Briefly, samples weighing 0.5 g were homogenized in 1.5 mL of ice-cold 0.05 mM 1,4-piperazinediethanesulphonic acid with pH of 7.0 and centrifuged (30 min, 12,850× *g*, 4 °C). Cholesterol and triacylglycerol concentrations were analyzed in the supernatant using the ready-to-use reagents (ELITech Group, Puteaux, France) and a MAXMAT PL biochemical analyzer (Erba Diagnostics France SARL, Montpellier, France).

### 4.14. Markers of Oxidative Stress

#### 4.14.1. Lipid Peroxidation Assay

The degree of lipid peroxidation in the cardiac muscle was determined spectrophotometrically using the TBARS (thiobarbituric-acid-reactive substance) assay [69]. The absorbance was measured at 532 nm using a Unicam UV300 spectrophotometer (Thermo-Spectronic, Cambridge, UK). TBARS concentration was calculated from a standard curve for malonyldialdehyde.

#### 4.14.2. Analysis of the Activity of DNA Repair Enzymes

The cardiac muscle samples were homogenized with 4 volumes of 50 mM Tris–HCl buffer (pH 7.5) containing 1 mM EDTA and protease inhibitor cocktail (Sigma-Aldrich, Darmstadt, Germany). Cells were disrupted through sonication (three 15 s pulses with 30 s intervals). The cell debris was removed via centrifugation (7000× *g*, 4 °C, 15 min) and the supernatant was used as the source of DNA repair enzymes. Protein concentration was determined using the Bradford method [76]. Supernatants were stored in aliquots at −80 °C for further analysis. Etheno adduct excision activity was measured using the nicking assay [34].

The nicking assay is based on the cleavage of the oligodeoxynucleotide at the site of modified bases (exocyclic DNA base adducts) such as 1,*N*^6^-ethenoadenine (etA), 3,*N*^4^-ethenocytosine (etC), and 8-oxo-deoxyguanosine (8-oxoG) by glycosylases and AP-endonucleases contained in tissue homogenates, as described previously [34]. The reaction mixture contained 5′-radiolabelled (^32^P) synthetic oligodeoxynucleotide duplexes (0.2 pmol) with a single DNA adduct (etA, etC, 8-oxoG), a reaction buffer, and incremental protein extracts (1 µg, 5 µg, 15 µg, 25 µg, 50 µg). Additionally, as a control for the oligodeoxynucleotide containing etA, etC, and 8-oxoG, a reaction without the tissue homogenate was performed. The reaction was carried out for 1 h at 37 °C. Since repair enzymes and some other proteins have a strong affinity for DNA, which interferes with the analysis of the products, after the end of the enzymatic reaction, the protein of the tissue extract was digested with proteinase K (20 μg/sample) for 1 h at 37 °C. The 8-oxoG oligomer was further cut via incubation in 0.2 N NaOH at 70 °C for 15 min. After the reaction, the stop solution with 95% formamide was added and samples were denatured for 3 min at 90 °C, then chilled on ice and briefly vortexed. The reaction products were separated into 20% polyacrylamide gels containing 7 M urea as a denaturing agent. The electrophoresis was carried out in 1× concentrated TBE buffer for 2–3 h at a voltage of 400–600 V (using a CONSORT 3000 V-300 mA high-voltage power supply). The gels were then autoradiographed for 12–16 h at −80 °C and the resulting autoradiographs were scanned on a Personal Laser Densitometer (Molecular Dynamics, Sunnyvale, CA, USA) and analyzed using Image Quant 5.2 and Microcal Origin software ver TL 10.2) The ratio of the cleaved to the total amount of ³²P-labelled oligonucleotide was calculated and presented in fM/h/µg protein units.

#### 4.14.3. Analysis of Gene Expression of DNA Repair Enzymes

The expression of 1 OGG1 (8-oxoguanine DNA glycosylase), MPG (N-Methylpurine DNA Glycosylase), TDG (Thymine-DNA glycosylase), APEX (apurinic/apirymidinic endodeoxyribonuclease 1), PTGS2 (prostaglandin-endoperoxide synthase 2), and ALOX (Arachidonate Lipoxygenase) genes was measured using Real-Time RT-PCR (qPCR). Briefly, total RNA was extracted from cardiac muscle using the PureLink RNA kit (Invitrogen, Waltham, MA, USA). RNA quantity and purity were determined using NanoDrop ND-1000 microspectrophotometer (Wilmington, DE, USA). A total of 1 μg of the total RNA was reverse-transcribed using Super Script IV Reverse Transcriptase (ThermoFisher Scientific, Waltham, MA, USA) and the mixture of random synthetic oligonucleotides (Applied Biosystems, Waltham, MA, USA). Real-Time PCR primers were designed with PrimerExpress software ver 3.0 (Table 2). The product from each pair was 131–140 bp long.

The primers were synthesized in DNA Sequencing and Oligonucleotide Synthesis Laboratory Institute of Biochemistry and Biophysics, Polish Academy of Sciences (IBB PAS). qPCR detection utilized SYBR Green I dye in combination with High-Capacity RNA-to-cDNA Kit Reverse Transcriptase (Applied Biosystems) using the real-time detection system CFX96 Dx (Bio-Rad, Hercules, CA, USA) according to manufacturer’s instructions. Optimized qPCR conditions were as follows: one cycle of 48 °C, 7 min; one cycle of 30 s at 95 °C; and 45 cycles of 20 s at 95 °C and 35 s at 62 °C, in which an optical acquirement was performed. Melting curve analysis for each reaction proved the absence of nonspecific products. Quantification was performed by the 2−ΔΔ*CT* method [55] using the 18S rRNA gene as the reference gene [34].

#### 4.14.4. Estimation of Oxidative Damage of Genomic DNA

Genomic DNA was isolated from frozen cardiac muscle samples using an AccuPrep Genomic DNA Extraction Kit (Cat. No. K-3032R, Bioneer Company, Seoul, Republic of Korea) according to the manufacturer’s protocol. Genomic DNA concentration and purity were determined on a NanoDrop ND-1000 spectrophotometer. Next, the DNA was digested by formamidopyrimidine-DNA glycosylase (Fpg, New England Biolabs (Ipswich, MA, USA), cat no. M0240S, 8000 U/mL) according to the manufacturer’s instructions. Control reaction omitted Fpg. After this step, nondigested and digested DNA samples were evaluated using electrophoresis on 1% agarose gel containing ethidium bromide. Upon completion of the electrophoresis, gels were photographed using a GelDoc-It Imaging System. Next, the optical density of DNA bands was analyzed using Image Quant Software ver TL 10.2 and the percent of digested DNA was calculated in relation to the total DNA.

#### 4.14.5. Statistical Analysis

All experimental data from at least three different trials (*n* = 3) were given in terms of mean standard error (SEM). To compare pairs of means, the Tukey post hoc test was used, indicating statistical significance with * *p* < 0.05, ** *p* < 0.1, and *** *p* < 0.01. For quantitative variables with a normal distribution, the Pearson correlation coefficient was selected. If the data were not normally distributed or had ordered categories, Kendall’s or Spearman’s tau-b coefficients, which are measures of connections between ranks, were used. Also, τ-b measures a monotonic relationship. Unlike Spearman’s ρ, we use it when the number of related ranks is large (i.e., when we have a small number of values for both variables—five or more). For genomic DNA oxidative damage data, the differences between the groups were analyzed using one-way analysis of variance (ANOVA) at the significance level of *p* = 0.05. Post hoc tests for pair-wise differences and the identification of homogeneous subgroups were conducted using Tukey’s HSD procedure. ANOVA was computed using Statistica 10 software (StatSoft Inc., Tulsa, OK, USA). Figures were prepared using Statistica 10 (scatter plots) and GraphPad Prism 5 software (remaining graphs).

## 5. Conclusions

It can be stated that in the early stage of atherosclerosis, lipid peroxidation, increased uric acid concentration, and VEGF-C (involved in the repair of damaged heart muscle) seem to be mechanisms that protect the genetic material against damage. These compensatory mechanisms and the oxidative stress markers OGG1, MPG, and ADMA (a marker of endothelial dysfunction) appear to be stimulated mainly by metabolic disturbances associated with a rapid change in diet, even if the change involves a return to a balanced diet. With the long-term use of an unbalanced diet, the levels of all DNA repair genes are increased, especially (significantly) Apex, Alox, and Ptgs, which strongly supports the hypothesis that an unbalanced diet induces oxidative stress that deregulates DNA repair mechanisms and may contribute to genome instability and tissue damage.

### Limitations

This study has potential limitations. It is important to remember that in our study, we used myocardial tissue composed of several cell types, not an isolated endothelial tissue. In the future, it would be valuable to confirm our findings using only endothelial tissue or with the use of immunostaining.

## Figures and Tables

**Figure 1 ijms-25-02282-f001:**
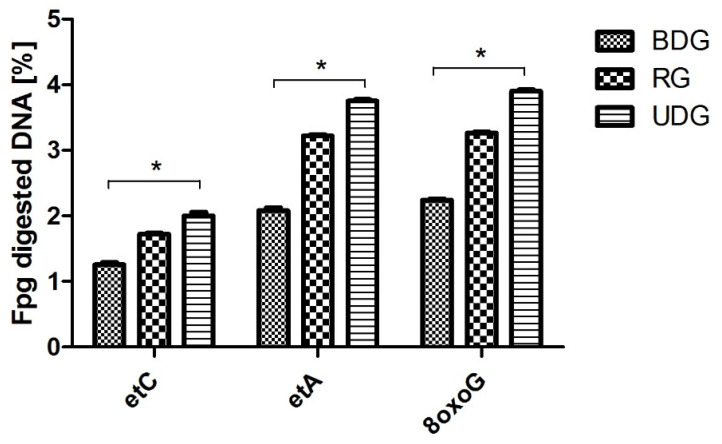
The activity of DNA repair enzymes (fmol/μg protein/h) in cardiac muscle of pigs in groups fed a balanced diet (BDG) and unbalanced diet (UDG) and pigs fed 9 months of unbalanced diet followed by 3 months of balanced diet—regression group (RG). *N*^4^-ethenocytosine (etC), N, 1,*N*^6^-ethenoadenine (etA), and 8-oxo-deoxyguanosine (8oxoG). Significance of differences for different type of diet was noted at * *p* < 0.05.

**Figure 2 ijms-25-02282-f002:**
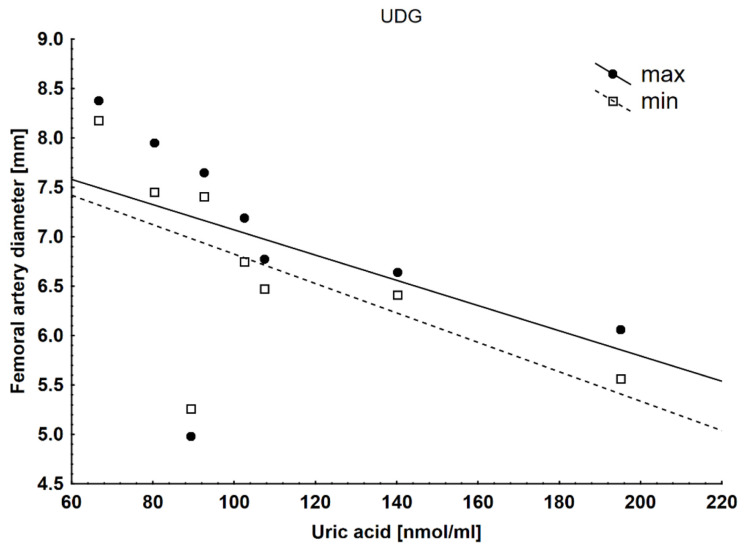
Negative correlations with the maximal (rs = −0.65) and minimal (rs = −0.59) diameter of the femoral artery in the UDG.

**Figure 3 ijms-25-02282-f003:**
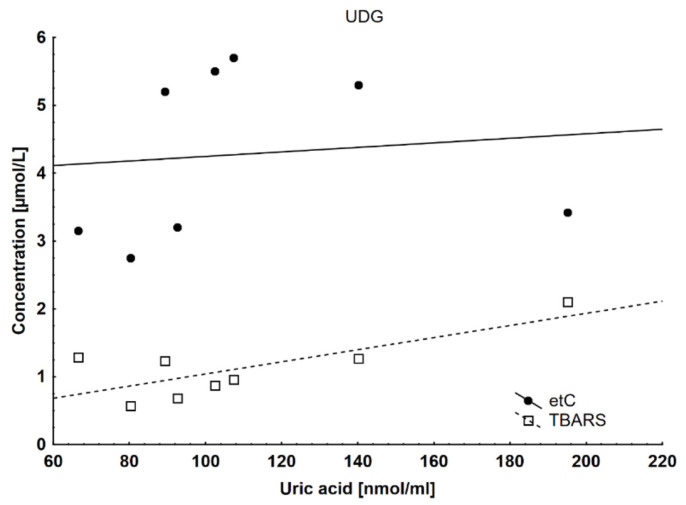
A positive correlation between uric acid and etC r_s_ = 0.5, and TBARS r_s_ = 0.62 in the UDG.

**Figure 4 ijms-25-02282-f004:**
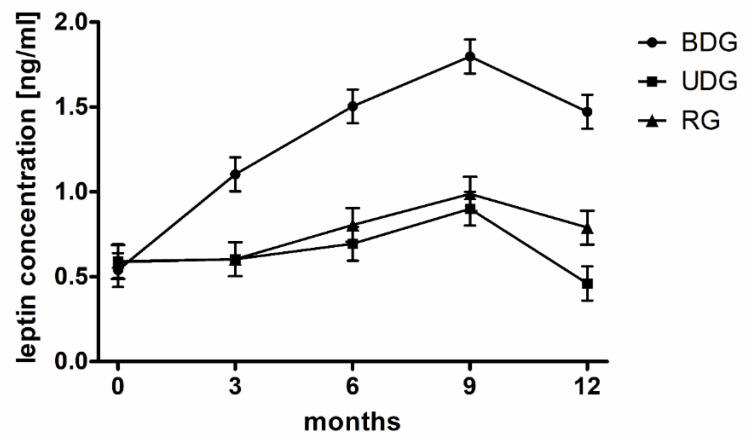
Leptin concentrations (ng/mL) in groups fed a balanced diet (BDG) and unbalanced diet (UDG) and in pigs fed 9 months of unbalanced diet followed by 3 months of balanced diet (regression group (RG)).

**Figure 5 ijms-25-02282-f005:**
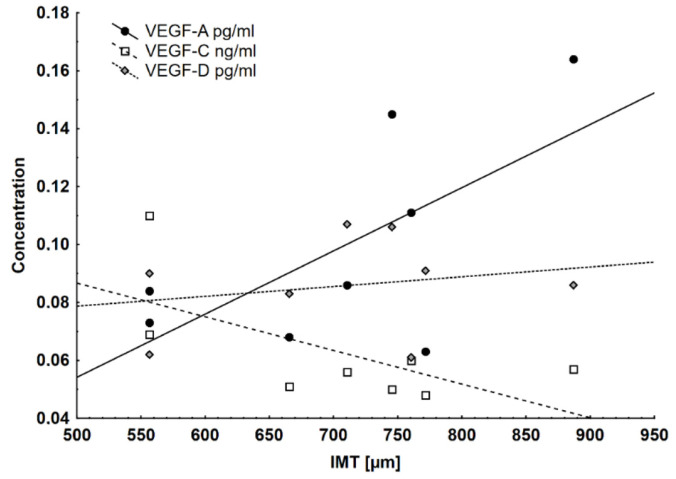
Serum VEGFs showed various associations with IMT FA. The positive correlation between IMT FA and VEGF-A (rs = 0.596) and the negative (insignificant) correlation between IMT and VEGF-C (rs = 0.52) were statistically significant.

**Figure 6 ijms-25-02282-f006:**
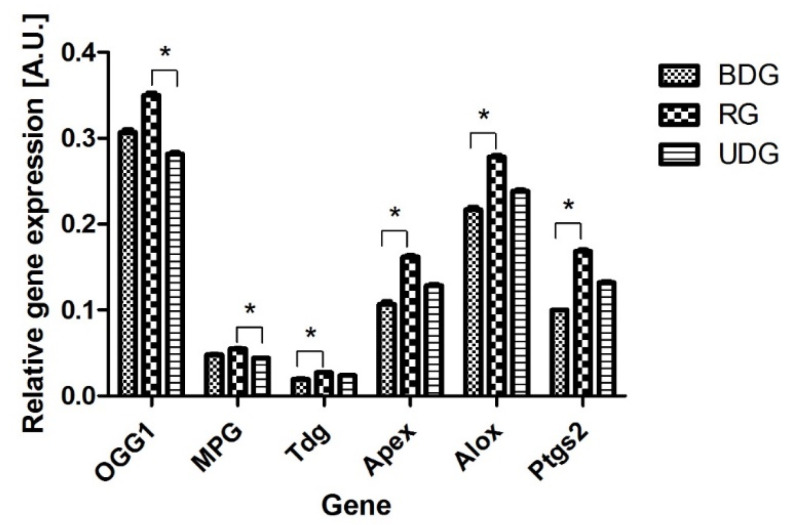
Expression of gene transcripts for all DNA repair system enzymes: MPG, TDG, OGG1, ALOX and APEX, and PTGS. * statistically significant at *p* < 0.05.

**Figure 7 ijms-25-02282-f007:**
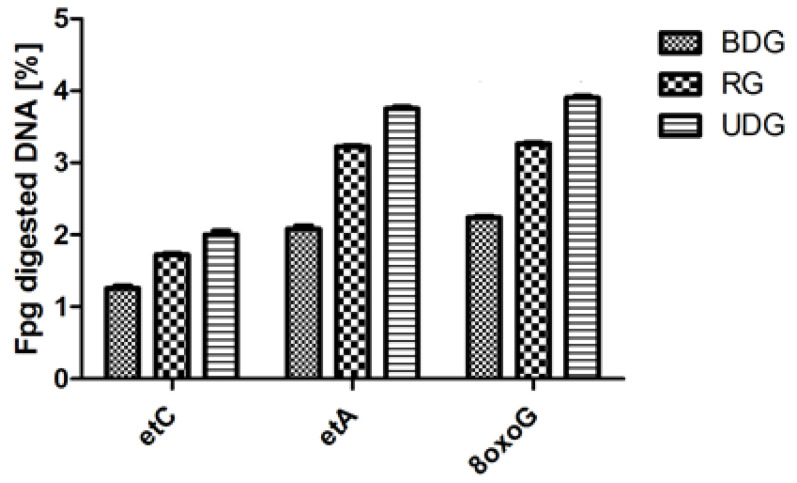
Percent of Fpg-digested genomic DNA isolated from cardiac muscle of pigs: RG (regression group), BDG (balanced diet group), and UDG (unbalanced diet group).

**Table 1 ijms-25-02282-t001:** Echocardiographic measurements and blood analyses in the three study groups at the start and the end (12 months) of the study. End-diastolic interventricular septum thickness (IVSd), the end-diastolic thickness of the free wall of the left ventricle (LWd), the thickness of complex intima + media of the femoral artery (IMT FA), maximal diameter of the femoral artery (D_max_ FA), white blood cells (WBCs), homeostasis model assessment of insulin resistance (HOMA), nonesterified fatty acid (NEFA)**,** C-reactive protein (CRP), triglycerides (TG), interleukin (IL), tumor necrosis factor (TNF), vascular endothelial growth factor (VEGF), L-arginine, asymmetric dimethylarginine (ADMA), and symmetric dimethylarginine (SDMA) are depicted. Appendix A, containing the data that were previously published [41,82], is available in the Appendix A.

	BDG		UDG		RG	
	Start	End	Start	End	Start	End
Body mass (kg)	40	246 ± 27.5	40	260 ± 21	40	245 ± 16.5
Blood pressure (mmHg)						
Systolic/	148 ± 10	153 ± 16	143 ± 14	161 ± 17	143 ± 14	159 ± 15
Diastolic	87 ± 11	98 ± 13	93 ± 13	103 ± 13	93 ± 13	109 ± 11
IVSd (mm)	0.78 ± 0.06	10.5 ± 0.9	0.78 ± 0.06	10.4 ± 0.9	0.78 ± 0.06	10.4 ± 0.9
LWd (mm)	0.7 ± 0.06	10.4 ± 0.7	0.7 ± 0.06	10.4 ± 0.5	0.7 ± 0.06	10.4 ± 10.6
D_max_ FA (mm)	5.07 ± 0.64	7.43 ± 1.18	5.18 ± 0.53	7.3 ± 1.05	5.18 ± 0.53	6.80 ± 0.88
WBC (G/L)	12.5 ± 2.96	10.82 ± 3.82	11.6 ± 2.77	9.47 ± 1.76	14.1 ± 2.62	9.87 ± 2.14
Urea (mmol/L)	3.35 ± 0.66	2.9 ± 0.78	6.62 ± 10.7	3.04 ± 1.04	3.76 ± 1.75	4.43 ± 1.37 ^
Creatinine (µmol/L)	182 ± 44	149 ± 24	165 ± 20	181 ± 44	180 ± 33	194 ± 30 ^
HOMA serum (Units)	0.21 ± 0.05	0.21 ± 0.06	0.25 ± 0.07	0.25 ± 0.07	0.35 ± 0.26	0.34 ± 0.25
NEFA plasma (mmol/L)	-	0.3 ± 0.15	-	0.29 ± 0.14	-	0.64 ± 0.44
CRP (ng/mL)	114 ± 46	60 ± 36	87 ± 43	52 ± 53	99 ± 33	41 ± 21
IL-1β (pg/L)	-	5.59 ± 6.08	-	3.89 ± 3.32	-	0.98 ± 1.59
IL-6 (pg/mL	-	4.52 ± 3.10	-	9.03 ± 3.03	-	9.70 ± 12.25
TNFα (pg/L)	-	59.0 ± 99.47	-	55.65 ± 65.86	-	36.50 ± 10.20
Leptin serum (ng/mL)	0.59 ± 1.38	1.48 ± 1.6	0.55 ± 0.47	0.47 ± 0.32 *	0.59 ± 0.45	0.83 ± 0.46
Uric acid serum (µmol/L)	111 ± 24	110 ± 24	103 ± 30	109 ± 41	111 ± 17	130 ± 24 ^□^
L-arginine—heart tissue (µmol/L)	74 ± 14	81 ± 10	76 ± 11	78 ± 13	76 ± 11	80 ± 12
ADMA heart tissue (µmol/L)	2.04 ± 0.25	1.37 ± 0.16 #	2.15 ± 0.1	1.46 ± 0.19 #	2.15 ± 0.1	1.5 ± 0.1 ^#
SDMA heart tissue (µmol/L)	0.73 ± 0.1	0.47 ± 0.07	0.8 ± 0.1	0.53 ± 0.07 *	0.7 ± 0.1	0.5 ± 0.08
VEGF-A serum (pg/mL)	0.11 ± 0.04	0.09 ± 0.02	0.09 ± 0.02	0.1 ± 0.04	0.11 ± 0.05	0.08 ± 0.02
VEGF-C serum (pg/mL)	0.07 ± 0.02	0.05 ± 0.01	0.08 ± 0.02	0.07 ± 0.02	0.07 ± 0.02	0.07 ± 0.02 ^
VEGF-D serum (pg/mL)	0.08 ± 0.02	0.09 ± 0.02	0.07 ± 0.01	0.08 ± 0.02	0.08 ± 0.02	0.08 ± 0.03
Cholesterol heart tissue (µmol/g)	-	0.57 ± 0.27	-	0.62 ± 0.16	-	0.71 ± 0.18
TG heart tissue (µmol/g)	-	6.68 ± 1.17	-	6.07 ± 0.2 *	-	5.63 ± 0.96 ^
TBARS heart (nmol/g)	-	0.89 ± 0.22	-	1.12 ± 0.48	-	1.19 ± 0.5
etC heart (fmol/μg protein/h)	-	4.81 ± 1.59	-	4.28 ± 1.25	-	5.94 ± 1.5^
etA heart (fmol/μg protein/h)	-	6.55 ± 2.84	-	6.16 ± 2.78	-	8.45 ± 2.2
8 oxoG heart (fmol/μg protein/h)	-	11.75 ± 2.67	-	11.05 ± 3.38	-	13.7 ± 2.5

* *p* < 0.05 for BDG vs. UDG; ^ *p* < 0.05 for BDG vs. RG; ^□^
*p* < 0.05 for UDG vs. RG; # *p* < 0.05 (start- vs. end-examination).

**Table 2 ijms-25-02282-t002:** Oligonucleotide primers used for Real-Time Quantitative RT-PCR analysis. A total of 140 bp of each product is shown.

Gene	Primer	Sequence (5′ to 3′)
*MPG*	MpgF	GTCCTAGTCCGGCGACTTCC
	MpgR	CTT GTCTGGGCAGGCCCTTTG C
*OGG1*	OGG1F	CTCAGAAATTCCAAGGTGTTC
	OGG1R	CCGCTCCACCAT-GCCAGTG.
*ALOX12*	Alox12F	TTGCATACTTTGTAGACAGTCTCC
	Alox12R	CTGAGTTTCATCCATTTTGGTCATG
*Ptgs2*	Ptgs2F	AAGGAGATGGCAGCAGAGTT
	Ptgs2R	GTGGCCGTCTTGACAATGTT
*18S rRNA*	18SpfF	ATCCTTCGATGTCGGCTCTT
	18SpfR	ACTAACCTGTCTCACGACGGTC
*ANPG*	ANPGpeF	CGCAGCATCTATTTCTCAAGC
	ANPGpeR	GTGCCATTAGGAAGTCGCC
*TDG*	TDGpeF	TAATGGGCAGTGGATGACCC
	TDGpeR	TGCAGCATTTAAGCAGAGCTGA
*APEX1*	APE1peF	GAATGCTGGCTTCACTCCACA
	APE1peR	AAAGGTGTAGGCATACGCCGT

## Data Availability

Data is contained within the article and Appendix A.

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
