# Peer review of "Analysis of the Model of Atherosclerosis Formation in Pig Hearts as a Result of Impaired Activity of DNA Repair Enzymes"

_ijms, 2024, doi:10.3390/ijms25042282_

Round 1
Reviewer 1 Report (New Reviewer)
Comments and Suggestions for Authors
This study demonstrates a better understanding of the biological influence of the modulations of a diet on the formation of atherosclerosis in pig hearts. The author claims that it results from the impaired activity of DNA repair enzymes. The topic is relevant and addresses a specific gap in the field of studies using animal models of the natural, slow development of atherosclerosis. It offers data that is not achievable in human subjects, particularly concerning the molecular alterations in heart tissue during the initial phases of atherosclerosis.
The introduction part should be rewritten. Additional information about atherosclerosis would be appropriate to add.
For further control, male pigs should be considered.
The author used ELISA, RT-PCR, ECHO, LC-MS, and other techniques to obtain the data.
A statistical program is advised to be changed (like GraphPad Prism) or major changes in the graphical appearance made. The descriptions of the figures could be more uniform and readable. The statistical significance should be added to the graphs.
The author showed the relevance of diet in the progression of vascular plaque and suggested DNA repair enzymes implicated in the pathophysiology of atherosclerosis. Unbalanced DNA repair that increases genomic instability and causes lesions to occur in tissues was demonstrated.
The references are appropriate.
It is a sloppy manuscript with many grammar mistakes, too complex sentences that are very hard to follow, and additional dots or dashes like lines 7, 9, 11, and 43.
Comments on the Quality of English LanguageIt is a sloppy manuscript with many grammar mistakes, too complex sentences that are very hard to follow, and additional dots or dashes like lines 7, 9, 11, and 43.
Author Response
Reviewer 1
Thank you very much for all the very valuable suggestions that contributed to improving the quality and substantive value of our manuscript
This study demonstrates a better understanding of the biological influence of the modulations of a diet on the formation of atherosclerosis in pig hearts. The author claims that it results from the impaired activity of DNA repair enzymes. The topic is relevant and addresses a specific gap in the field of studies using animal models of the natural, slow development of atherosclerosis. It offers data that is not achievable in human subjects, particularly concerning the molecular alterations in heart tissue during the initial phases of atherosclerosis.
1.The introduction part should be rewritten. Additional information about atherosclerosis would be appropriate to add.
1.Answer:
The introduction part has been rewritten.
- For further control, male pigs should be considered.
Answer:
The experiment was aimed at minimizing additional disturbing factors, including the activity of sex hormones, and the fight for dominance. Therefore, the experiment was conducted on non-pregnant sows, most often during the apoestrous period. Therefore, the experiment was carried out on non-pregnant sows, almost all the time during the apoestrous period. Conducting experiments on males in practice turned out to be difficult for several reasons - unneutered males are aggressive toward each other - and male piglets are castrated in the first days of life for ethical reasons. If an experiment is planned on males, the breeder must be informed not to castrate a certain number of males. However, in such a case, either the males would not have identical body weights when included in the experiment, or a very large group of boars would have to be left to enable the selection of animals with identical body weights.
A statistical program is advised to be changed (like GraphPad Prism) or major changes in the graphical appearance made.
We changed the data visualization software to standardize and transparently present the data in all figures. We included the software information in the Materials and Methods section.
The descriptions of the figures could be more uniform and readable. The statistical significance should be added to the graphs.
We corrected the figure description.The statistical significance has been included in the appropriate figures illustrating the data
It is a sloppy manuscript with many grammar mistakes, too complex sentences that are very hard to follow, and additional dots or dashes like lines 7, 9, 11, and 43.
The manuscript was corrected both grammatically and stylistically.
Comments on the Quality of English Language
The language has been improved.
Reviewer 2 Report (New Reviewer)
Comments and Suggestions for Authors
The intent of the manuscript "Analysis of the model of atherosclerosis formation in pig hearts as a result of impaired activity of DNA repair enzymes" is to attempt to establish an association of endothelial dysfunction markers and repair enzyme activity in a swine atherosclerosis model. The topic is of prime interest, as atherosclerosis leads to more deaths than any other disease, but mechanistic insight of this condition is still lacking. The use of pigs is appreciated as well, since atherosclerotic development in swine is thought to largely resemble atherosclerosis in humans. However, there are several concerns with this manuscript, with the primary reasons being the weak end-points chosen along with lack of robust data present. These issues, along with no mechanistic data, significantly dampens the enthusiasm of this manuscript. Major reviewer comments are provided below:
*The manuscript lacks rigor due to being primarily descriptive in nature. If the end-points of interest really do influence atherosclerotic progression, there are no follow-up studies to directly test this, making data interpretation challenging as to whether measured markers do impact atherogenesis and atherosclerotic progression, and if so, how.
*The authors fail to confidently confirm atherosclerotic plaque within their models. Since swine were used and it is easy to harvest atherosclerotic tissues from these models, this is disappointing. Please include H&E, ORO, etc. to quantify the degree of atherosclerotic lesion area and lipid content in the pig groups.
*The reviewer assumes that the material used in the "Measurements of Insulin, Leptin, Vascular Endothelial Growth Factor (VEGF), C-reactive 519 protein (CRP), Uric Acid, and Glucose" analyses was serum/plasma, but this is not well defined. Please clarify by being more explicit.
*Why were uneven groups used in the study? Were some animals omitted for downstream analyses, and if so, why? Please clearly explain the rationale of group numbers used/chosen. Also, to enhance rigor, male pigs should have also been included in the study, making this a limitation. Please explain why only females were utilized.
*The authors appear to make conclusions (e.g. endothelial dysfunction) that do not seem to be supported by their data. For instance, material used for some end-point measurements was myocardium & cardiac muscle, when more rigorous analyses would include actually assessing the endothelium via immunostaining to directly assess endothelial dysfunction. Please include more relevant end-points to support study conclusions, or soften the overall tone of data interpretation based on current manuscript findings. Most of the markers measured are either systemic or from cardiac tissue, which leaves a wide interpretation as to exactly why these markers are impacted. Honing in on certain cells of the artery and making more precise measurements would strengthen the manuscript.
*Overall, the analyses used by the authors is poor, especially when taking into consideration the large amount of material available from the models employed. Protein expression should be included with gene expression, along with immunostaining of sectioned arteries to provide more robust data.
*The Discussion and Conclusions section reads more like a review article and it is recommended to overhaul this section by succinctly focusing on the manuscript data, giving the backdrop of the current literature and how the manuscript may aid the scientific community, study limitations, and conclusions and future directions.
Comments on the Quality of English LanguageUse of English language needs significant improvement to be publication-ready.
Author Response
Reviewer 2
Thank you very much for all the extremely valuable suggestions that contributed to improving the quality and substantive value of our manuscript
The intent of the manuscript "Analysis of the model of atherosclerosis formation in pig hearts as a result of impaired activity of DNA repair enzymes" is to attempt to establish an association of endothelial dysfunction markers and repair enzyme activity in a swine atherosclerosis model. The topic is of prime interest, as atherosclerosis leads to more deaths than any other disease, but mechanistic insight of this condition is still lacking. The use of pigs is appreciated as well, since atherosclerotic development in swine is thought to largely resemble atherosclerosis in humans. However, there are several concerns with this manuscript, with the primary reasons being the weak end-points chosen along with lack of robust data present. These issues, along with no mechanistic data, significantly dampens the enthusiasm of this manuscript. Major reviewer comments are provided below:
*The manuscript lacks rigor due to being primarily descriptive in nature. If the end-points of interest really do influence atherosclerotic progression, there are no follow-up studies to directly test this, making data interpretation challenging as to whether measured markers do impact atherogenesis and atherosclerotic progression, and if so, how.
Answer:
Our study focused on the analysis of oxidative stress induced by specific dietary patterns, investigating the formation of modified DNA bases. The nicking assay method was employed to assess these modifications, with a specific focus on the involvement of DNA repair enzymes within the Base Excision Repair (BER) pathway.
Given the nature of our study, it's important to note that follow-up studies, as commonly conducted in human studies, are less prevalent in pig animal model. Nevertheless, we acknowledge your concern about the need for more direct tests on the influence of our end-points on atherosclerotic progression. We will consider addressing this in future research by studying the specific impact of the measured markers on atherogenesis and atherosclerotic progression.
*The authors fail to confidently confirm atherosclerotic plaque within their models. Since swine were used and it is easy to harvest atherosclerotic tissues from these models, this is disappointing. Please include H&E, ORO, etc. to quantify the degree of atherosclerotic lesion area and lipid content in the pig groups.
Answer:
The authors undoubtedly confirmed the presence of atherosclerotic plaques in the artery walls in histopathological examinations. We have included data on this subject in our previous publications.
*The reviewer assumes that the material used in the "Measurements of Insulin, Leptin, Vascular Endothelial Growth Factor (VEGF), C-reactive 519 protein (CRP), Uric Acid, and Glucose" analyses was serum/plasma, but this is not well defined. Please clarify by being more explicit.
Thank you for this comment. The material used in the analyzes "Measurements of insulin, leptin, vascular endothelial growth factor (VEGF), C-reactive protein 519 (CRP), uric acid and glucose" was serum. We added this information and marked it green in the text and in the Table 1. with the results. Now it is precisely exposed in the methodological part. Another results as given in supplementary part with reviewer suggestions.
*Why were uneven groups used in the study? Were some animals omitted for downstream analyses, and if so, why? Please clearly explain the rationale of group numbers used/chosen. Also, to enhance rigor, male pigs should have also been included in the study, making this a limitation. Please explain why only females were utilized.
Initially, all pig groups consisted of 11 pigs. However, five pigs developed renal cysts and were eliminated, even though all blood tests were within normal limits. These pigs were euthanized, and cysts were confirmed by postmortem examination. Observations on 2013 in the article Pasławski R., Janiszewski A., Noszczyk-Nowak A., Nowacki D., Pasławska U.: Polycystic kidney disease in White House Pigs. EJPAU 2013, 16(2), #4
*The authors appear to make conclusions (e.g. endothelial dysfunction) that do not seem to be supported by their data. For instance, material used for some end-point measurements was myocardium & cardiac muscle, when more rigorous analyses would include actually assessing the endothelium via immunostaining to directly assess endothelial dysfunction. Please include more relevant end-points to support study conclusions, or soften the overall tone of data interpretation based on current manuscript findings. Most of the markers measured are either systemic or from cardiac tissue, which leaves a wide interpretation as to exactly why these markers are impacted. Honing in on certain cells of the artery and making more precise measurements would strengthen the manuscript.
Regarding the interpretation of our data, we understand your suggestion to soften the overall tone and better adapt our conclusions to the available data. We understand that more rigorous analyses require isolated endothelial tissue or immunostaining. Within our financial possibilities, we chose ADMA (with the addition of SDMA and L-arginine), blood pressure measurement, and ultrasound measurements of the ability to dilate arterial walls as an expression of endothelial function. We are aware of the limitations of such a single measurement. A section of the heart containing a quite large diameter coronary artery (second branch) was taken for analysis. We treated the myocardium and endothelium as one whole.
*Overall, the analyses used by the authors is poor, especially when taking into consideration the large amount of material available from the models employed. Protein expression should be included with gene expression, along with immunostaining of sectioned arteries to provide more robust data.
Protein expression analysis studies relating to the DNA repair enzymes of the BER pathway were included in the results and for nicking assay activity for DNA damage-induced ethenoadenine, ethenocytosine and 8oxoguanine. All these damages are recognized by Mpg, Tdg and OGG1 enzymes, which expressions were measured(see discussion part)
*The Discussion and Conclusions section reads more like a review article and it is recommended to overhaul this section by succinctly focusing on the manuscript data, giving the backdrop of the current literature and how the manuscript may aid the scientific community, study limitations, and conclusions and future directions.
The discussions and conclusions have been corected.
Comments on the Quality of English Language
Use of English language needs significant improvement to be publication-ready.
Language correction has been performed
Reviewer 3 Report (New Reviewer)
Comments and Suggestions for Authors
Review Comments
In this article, the authors studied the role of DNA repair enzyme activity in the atherosclerosis formation in pig hearts. The increased level of oxidative stress in UDG pig hearts is associated with a high level of DNA repair enzymes.
Major comments:
In Table 1: column RG, the end body mass 24.5 should be changed to 245.
The authors have already published some of the results from Table 1 in the following publication, Adam Zabek et al., PLOS ONE | https://doi.org/10.1371/journal.pone.0184798.
Table 1 is too busy. Hence, remove the results which do show any difference between the groups and provide a table in the supplementary file. In main text, keep a table with the parameters which show difference between groups.
This study clearly shows that the oxidative stress plays an important role in the early stages of atherosclerosis formation.
Author Response
Reviewer 3
Thank you very much for all the extremely valuable suggestions that contributed to improving the quality and substantive value of our manuscript
In this article, the authors studied the role of DNA repair enzyme activity in the atherosclerosis formation in pig hearts. The increased level of oxidative stress in UDG pig hearts is associated with a high level of DNA repair enzymes.
Major comments:
In Table 1: column RG, the end body mass 24.5 should be changed to 245.
Thank you for this comment. The value has been changed.
The authors have already published some of the results from Table 1 in the following
publication, Adam Zabek et al., PLOS ONE | https://doi.org/10.1371/journal.pone.0184798.
To perform a comparative analysis with new results regarding the analysis of damage and repair in the BER system using the nicking assay method, which is a perfect complement to this type of research the whole then constitutes a coherent element
Table 1 is too busy. Hence, remove the results that do not show any difference between the groups and provide a table in the supplementary file. In the main text, keep a table with the parameters that show differences between groups.
Table 1 has been split. Table 1 now shows only the new results. Previously published data are summarized in another table included in the Supplementary Data. We believe that now Table 1 will be clearer, and readers will be more comfortable reading the article when the most important data we refer to will be at hand in the supplementary data.
Round 2
Reviewer 2 Report (New Reviewer)
Comments and Suggestions for Authors
When assessing the author comments and their revisions, there is a pressing question from the reviewer regarding whether these animals have been used in prior published studies, and now the scientists are utilizing specimens from these animals to measure different end-points to be used in a separate publication (this currently reviewed manuscript). If this is the case, then this needs to be made very explicit, and appropriate citations added/included.
If this is not the case, then confirmation of atherosclerosis needs to be confirmed, along with appropriate quantification of atherosclerotic lesion area and lipid content via H&E and ORO stains, respectively. Just because a dietary protocol that has been used is known to induce atherosclerosis in their respective animal model, scientists should make the assumption that extensive atherosclerotic progression has occurred. Instead, this should be shown with high-quality data.
Comments on the Quality of English LanguageThere still should be substantial editing in regards to the English language used in the manuscript.
Author Response
Answer to Reviewer 2
Thank you very much for your extremely accurate statement. Referring to this, we state that:
The authors wrote in the materials and methods section that after 12 months of the experiment, the pigs were euthanized, and immediately after each animal was euthanized, samples of various organs were taken from it and stored for later laboratory tests. We described in detail how the samples were secured for further testing. It is also clearly noted that in previous articles we presented evidence of metabolic changes in blood and tissue tests and atherosclerotic changes in arteries in pigs fed improperly (UDG). The development of atherosclerotic lesions was confirmed by changes in blood tests (Tables 1 and 2), as well as the results of ultrasound examination (increase in blood pressure, decrease in the diameter of the external carotid artery in diastole and systole measured using the e-tracking method, i.e. averaged measurement from many pulse cycles, an increase in IMT thickness in ultrasound) and histopathological examination. The presence of atherosclerotic plaques in the artery walls was confirmed histopathologically and changes in gene expression were demonstrated in the artery walls of UDG and RG pigs. In the authors' opinion, this evidence is sufficient to conclude that the applied animal experimental model allowed to obtain changes very similar to early human atherosclerosis. After discussing the reviewer's comments with the team of researchers, we propose not to make any changes in this part of the manuscript. If our explanations in the article seem to be insufficient for the reviewer, please indicate the place (sentence) , which is unclear according to the reviewer.
Round 3
Reviewer 2 Report (New Reviewer)
Comments and Suggestions for Authors
When assessing the R2 Response to Reviewers comments, a claim is made regarding the development of atherosclerotic lesions being confirmed with blood tests. I don't think so, as no blood test can confirm atherosclerosis. The same goes for blood pressure measurements. These are just risk factors. The only data of atherosclerosis is IMT measurements, but this is indirect evidence. The authors mention histopathological examination in their rebuttal, but I don't see anything related to that in the manuscript.
Atherosclerosis is analyzed by measuring atherosclerotic lesion area and lipid content. Unfortunately, the manuscript does not show this and the prior studies that are referenced in the supplementary Table S1 [41, 82] don't look correct, based on the cites in the main text.
Confirming atherosclerotic lesions are present is simply, straightforward, and will confirm atherosclerosis is present in your models. Below are manuscripts which justify this and these approaches should be utilized to confirm atherosclerosis with direct evidence.
https://pubmed.ncbi.nlm.nih.gov/28729366/
https://pubmed.ncbi.nlm.nih.gov/8274468/
https://pubmed.ncbi.nlm.nih.gov/11116057/
Comments on the Quality of English LanguageUse of English language needs to be vastly improved.
This manuscript is a resubmission of an earlier submission. The following is a list of the peer review reports and author responses from that submission.
Round 1
Reviewer 1 Report
Comments and Suggestions for Authors
The manuscript submitted by Paslawski et al. aimed to evaluate the effect of high calorie diet on various markers of oxidative stress and cardiovascular function in association with DNA repair enzymes.
The main topic of this study could be interesting and relevant but the manuscript has many weaknesses and inaccuracies. Methods and results are not properly described. Discussion is too long and the conclusion is not consistent with the title. Overall, the manuscript is in need of correction and further requires extensive English editing.
Major comments:
1- The authors stated in the title that ‘Endothelial Dysfunction and the Impaired Activity of DNA Repair Enzymes in the Heart is Present in Very Early Stages of Arteriosclerosis’. The major problem is that nothing in this study allows to highlight endothelial dysfunction or early atherosclerosis in this swine model. Endothelial function was not studied and endothelial dysfunction was not evidenced. Additionally, no argument or results established that this model is a model of early atherosclerosis. This needs to be explained or explored further.
2- The authors performed 12 months follow up of the animals with quarterly analyzes. It would have been relevant to show the evolution of the different parameters for each group over time. In particular, in the RG group, what is the effect of the change in diet on the development of metabolic syndrome in the animals?
3- RG group was fed a western type diet, is this diet high calorie? This is not clear.
4- The authors described in methods §4.5 Blood examination: total cholesterol and triglycerides were determined in blood serum. In the results, table 1 triglycerides and cholesterol levels seemed to be determined in the heart?
5- Some correlations are shown for uric acid and various parameters in UDG, how are these correlations in the other groups? What is the scientific relevance of such correlations?
6- All figures and table needs to be correctly presented. In almost all graphs there are no units, no legend and in figure 6 no explanation in the text. Moreover, Figure 6 is incomprehensible.
7- The statistical analysis method was not described in the study.
Minor comments:
1- Correct all along the text “hypertriglyceridemia” and not hipertriglicerolemia
2- Please show the unit in all graphs.
3- In table 1 a lot of typos. Problem with the data: IMT FA is not shown, heart triglycerides and cholesterol instead of serum triglycerides and cholesterol, body mass in RG end …
4- In figure 1, legend and symbols-colors do not correspond
5- In figure 3: no explanation is given on statistical analysis and the type of correlation.
6- Figure 4: what is the x-axis?
7- Figure 6: what is the color code. This is not specified either in the legend or in the text.
Comments on the Quality of English LanguageThe quality of english language is low and the manuscript needs english editing.
Author Response
Reviewer 1
Please find enclosed for your consideration a revised article
Firstly, we would like to express our gratitude to Reviewer 1 for all suggestions that allowed us to considerably improve our manuscript. To respond to the editor and reviewer queries we introduced a number of changes in the text (marked green colour). We have revised the text according to the suggestions and hope that you will now find it suitable for publication in IJMS journal.
Below, please find the detailed information on the changes in the manuscript with answers to all comments
Major comments:
- The authors stated in the title that ‘Endothelial Dysfunction and the Impaired Activity of DNA Repair Enzymes in the Heart is Present in Very Early Stages of Arteriosclerosis’. The major problem is that nothing in this study allows to highlight endothelial dysfunction or early atherosclerosis in this swine model. Endothelial function was not studied and endothelial dysfunction was not evidenced. Additionally, no argument or results established that this model is a model of early atherosclerosis. This needs to be explained or explored further.
The title of the mancucript has been changed on Analysis of the model of atherosclerosis formation in pig hearts as a result of impaired activity of DNA repair enzymes.
- The authors performed 12 months follow up of the animals with quarterly analyzes. It would have been relevant to show the evolution of the different parameters for each group over time. In particular, in the RG group, what is the effect of the change in diet on the development of metabolic syndrome in the animals?
Disruptions in the fetal environment can have serious consequences for offspring in adulthood. Despite the complex nature of metabolic disorders, studies on animal models of fetal growth restriction and excessive growth reveal several important mechanisms underlying the developmental programming of adult diseases. The use of animal models of disturbed fetal growth allows the identification of both structural and functional changes. In a disturbed intrauterine environment, development passes through successive critical points, but the flexibility of changes in organ structure and their subsequent homeostatic functions become more limited. Changes in gene expression may persist throughout life, both as a result of remodeling of tissues and organs in the fetus and newborn, and in response to environmental stimuli after birth. This has been well documented in insulin resistance observed in many models of restricted fetal growth and in human studies. Insulin acts on various tissues in the body, but skeletal muscles, liver and adipose tissue are particularly important in regulating glucose homeostasis. Recent research shows that early programming of metabolic disorders is related to maternal nutrition experienced in early and later pregnancy. Each of these exposures acts through different mechanisms, such as changes in the efficiency of triglyceride deposition in the postnatal period, activation of the hormonal "stress axis" and its potential impact on glucose tolerance in the offspring. This is important in nutritional recommendations for future mothers who are overweight or obese, because any nutritional intervention in the period around conception may cause metabolic and endocrine consequences in the offspring.
- RG group was fed a western type diet, is this diet high calorie? This is not clear.
The Western diet is characterized by, among others: low in dietary fiber and high in saturated fat, simple sugar and fatty dairy products. However, such a diet lacks vegetables and fruit, low-fat dairy products and meat, as well as whole grain products. The Western diet is therefore high in energy and rich in fat, including saturated fatty acids, simple sugars and salt. It is often lacking in vitamins, minerals and fiber.
- The authors described in methods §4.5 Blood examination: total cholesterol and triglycerides were determined in blood serum. In the results, table 1 triglycerides and cholesterol levels seemed to be determined in the heart?
serum from blood from the heart. This has already been corrected in the table
- Some correlations are shown for uric acid and various parameters in UDG, how are these correlations in the other groups? What is the scientific relevance of such correlations?
These parameters for the UDG diet are much lower in other groups and are important in metabolic processes where the accelerated process of lipid peroxidation and oxidation of cellular components takes place.
Comparison of the results of the final study showed the highest concentration of ADMA in RG, while we expected this in UDG. This observation is consistent with our previous metabolomics studies conducted on the same pigs, in which also the greatest metabolic differences compared to the control group were found in RG [41]. SDMA is a structural isomer of ADMA and does not inhibit NOS [52]. Its known that SDMA is competing with l-arginine for transport across cell membranes [63]. There are also growing number of evidence supporting participation of SDMA in the development of inflammation and atherosclerosis [64]. We also observed the highest level of SDMA in the UDG (Table 1)
- All figures and table needs to be correctly presented. In almost all graphs there are no units, no legend and in figure 6 no explanation in the text. Moreover, Figure 6 is incomprehensible.
all drawings have been supplemented with missing descriptions
All full description is given upper on the Figure 6
Expression of gene transcripts for all DNA repair system enzymes (MPG, TDG, OGG1, ALOX and APE1, PTGS2) were found in all analyzed tissues (Figure 6.). Differ-ences in the number of analyzed transcripts between individual tissues from the con-trol and study groups are presented in (Figure 6) Interestingly, higher (p <0.05) mRNA levels for genes. In all analyzed cases, the mRNA level of gene expression in the control tissues was higher than in the heart tissues with regression in the diet. The observed levels of gene expression were the highest for the Apex gene and comparable with the mRNA values for the OGG1, ALOX, and PTGS2 genes. These values were 2 times high-er in relation to the Mpg and Tdg genes. Also, the mRNA level of the Mpg gene was 3 times higher than the mRNA of the Tdg gene. In the analyzed regression samples, in all analyzed cases the mRNA level was 4 times lower than in the control samples. All re-sults were statistically significant at the level of (p <0.05 - p <0.01), [54-56].
- The statistical analysis method was not described in the study.
Chapter 4.14.5. Statistical analysis
All experimental data from at least three different trials (n = 3) were given as means standard error (SEM, manufacturer, Saint Louis, MO, USA). To compare pairs of means, the Tukey post hoc test was used, indicating statistical significance with * p < 0.05, ** p < 0.1, and *** p < 0.01. For quantitative variables with a normal distribution, the Pearson correlation coefficient was selected. If the data were not normally distrib-uted or had ordered categories, Kendall's or Spearman's tau-b coefficients were used, which are a measure of connections between ranks. Also, τ-b measures a monotonic relationship. Unlike Spearman's ρ, we use it when the number of related ranks is large (i.e. when we have a small number of values of both variables - 5 or more).
Minor comments:
- Correct all along the text “hypertriglyceridemia” and not hipertriglicerolemia
nomenclature has been corrected
- Please show the unit in all graphs.
nomenclature in unit in all graphs has been corrected
- In table 1 a lot of typos. Problem with the data: IMT FA is not shown, heart triglycerides and cholesterol instead of serum triglycerides and cholesterol, body mass in RG end …
All typos has been changed and correct
- In figure 1, legend and symbols-colors do not correspond
Now has been corrected
- In figure 3: no explanation is given on statistical analysis and the type of correlation.
The explanation has been corrected.
- Figure 4: what is the x-axis?
On figure 4 the caption has been added under the drawing: balanced diet
- Figure 6: what is the color code. This is not specified either in the legend or in the text.
the drawing has been changed and corrected for better readability
Reviewer 2 Report
Comments and Suggestions for Authors
I congratulate the authors on a well-designed study, well-presented results and good discussion.
However, I would suggest that they devote a little more attention to the long-term risk factors that lead to vascular changes and how long it might take for these changes to occur. Biochemical parameters that change in a relatively short time are presented. What is the ratio of pro- to anti-inflammatory cytokines? is this result informative? Is there a possibility to measure the FMD of the vascular walls during the ultrasound measurement?
Gene expression analyses are included in the results, in great detail when it comes to 8oxoG, but for the others I suggest a few more additions (modified nucleosides).
Please include the strengths and weaknesses of your study
Author Response
Reviewer 2
Please find enclosed for your consideration a revised article
Firstly, we would like to express our gratitude to Reviewer 1 for all suggestions that allowed us to considerably improve our manuscript. To respond to the editor and reviewer queries we introduced a number of changes in the text (marked green colour). We have revised the text according to the suggestions and hope that you will now find it suitable for publication in IJMS journal.
Below, please find the detailed information on the changes in the manuscript with answers to all comments
What is the ratio of pro- to anti-inflammatory cytokines? is this result informative?
Atherosclerosis is considered a progressive inflammatory systemic disease affecting mainly the walls of the aorta, carotid, and coronary arteries with a long latency period [57,58]. Such a slow inflammatory process is not detected in standard tests: the animals had normal body temperature, leukocyte count, CRP, no differences in the levels of pro-inflammatory cytokines IL-1, IL-6, TNFα, and no inflammatory infiltrates in the artery walls [39]. However, analyzing previous research, this incomplete metabolic syndrome appears to be typical of the pig model
Is there a possibility to measure the FMD of the vascular walls during the ultrasound measurement?
To be assessed the flow reserve in the anterior descending branch of the left coronary artery in response to stressful stimuli echocardiography is performed. Another tool for assessing reserves changes in myocardial perfusion in response to adeno- zine or dipyridamole or stress while trying to cool is positron emission tomography. Recently to Magnetic resonance imaging was also used for this purpose heart, which allows assessment of the flow response to the entire thickness of the myocardial wall (in layers subendocardium and subepicardium). Intra- pellicle is, however, a systemic disorder, therefore, less invasive and cheaper techniques have emerged assessment of peripheral vascular endothelial function. An example of such a technique is ultrasound imaging of the brachial artery enabling measurements the size of the growth-dependent dilation of this artery blood flow (flow-mediated dilatation, FMD) caused by the occlusion of this artery for several minutes and subsequent release of the clamp, which induces postischemic dilation of small blood vessels supporting hands and forearms. Recently even described paradoxical contractile response to the release of the pressure of the brachial artery [83], the authors of this study found that this answer allows for prediction worse prognosis in women [84].
FMD measurement technique using ultrasound requires practice to minimize measurement variability. However, it is the most common non-invasive method of assessing function endothelium.
Questions”
Gene expression analyses are included in the results, in great detail when it comes to 8oxoG, but for the others I suggest a few more additions (modified nucleosides).
Please include the strengths and weaknesses of your study:
Answer:
Understanding the genotoxic properties of endogenous DNA damage as etheno DNA adducts, such as 1,N6-ethenoadenine (etA), 3,N4-ethenocytosine (etC), N2,3-ethenoguanine (etG) and 1,N2-ethenoguanine (etG) and repair pathways, resulting in the so-called oxidative stress process, is fundamental to understanding the mechanisms of diseases that depend on chronic inflammation such as cancer or neurodegenerative diseases, and aging processes. Due to the large range of these products and pleiotropic action, knowledge about the molecular mechanisms of their action is still fragmentary.
Among the most genotoxic oxidative DNA damages are 1, N6-ethenoadenine (eA), 3, N4-ethenocytosine (eC) and 8-oxoguanine (8-oxoG). All these lesions have a high miscoding potential and are found in DNA of untreated animals and humans. The level of ethenoadducts in DNA increases in Wilson disease and primary hemochromatosis, diseases leading to the development of cancer, as well as in ulcerative colitis and polyposis, suggesting their causative effect on cancer. 8-oxoG level was found to be increased in blood leukocytes of lung cancer patients in comparison with healthy controls due to decreased repair capacity. Decreased repair of eA and eC was also shown to accompany the development of a specific histological type of lung cancer, adenocarcinoma, the ethiology of which is linked to chronic inflammations and healing of scars.
8-oxoG as well as eA and eC are eliminated from DNA by Base Excision Repair (BER) pathway, initiated by DNA glycosylases, which cleave the N-glycosidic bond between the damaged base and the deoxyribose moiety, leaving behind a baseless sugar (AP-site). AP-sites are then processed by AP-endonuclease. DNA glycosylases excising 8-oxoG (OGG1) and eC (TDG) have a high affinity for reaction product, and do not leave the AP-site without the assistance of AP-endonuclease [19-20]. Ethenoadenine is excised from DNA by N-methylpurine-DNA glycosylase (MPG), which also repairs a broad spectrum of DNA alkylation products like 3-methyladenine or 7-methylguanine , hypoxanthine, ethylated bases , and 1,N2-etG. Thymine-DNA glycosylase (TDG) excises εC and thymine from G:T pairs resulting from deamination of 5-methylcytosine. OGG1 glycosylase, in addition to 8-oxoG, removes also unsubstituted and substituted imidazole ring-opened purines. OGG1 is endowed with AP-lyase activity and cleaves AP-sites by b-elimination. The major human 5’AP-endonuclease is APE1 (also called HAP-1/APX/REF1). Besides its AP-endonuclease activity, APE1 also acts as a redox regulation enzyme for transcription factor, as well as exhibits other activities, like 3’-5’exonuclease, phosphodiesterase, 3’phosphatase and RNase H.
Since DNA repair is an important risk factor in the pathogenesis of certain diseases, ongoing research is focused on factors which may modulate the rate of DNA repair. One of important modulators is oxidative stress. On the one hand, ROS can affect transcription of some repair enzymes, on the other, repair enzymes may be directly inactivated by ROS and RNOS. For example, exogenous nitric oxide and peroxynitrite have been shown to inhibit OGG1, DNA ligase formamidopyrimidine-DNA-glycosylase and O6-alkylguanine-DNA-alkyltransferase by direct nitrosylation .
OGG1, a housekeeping gene, was shown to be induced by oxidative stress may stimulate nuclear mobilization of Ogg1 protein in pig hepatocytes 2 h after treatment, although this response was delayed in 24-months old animals to 6 h, most probably due to a defect in nuclear import of Ogg1-a, which was abundant in the cytoplasm of aged cells. Thus, oxidative stress including with diet after their administration mobilizes the entry of existing Ogg1 molecules from the cytoplasm to the nucleus, and later may also stimulate its synthesis de novo. Mpg is regulated both in the cell cycle, being induced at the end of G1 and in the S phase, and by different exogenous factors, like ionizing radiation and other ROS generating factors. In our study transcription of Tdg was also induced moderately. The biggest induction after OGG1 the expression level was observed for AP-endonuclease (Ape1). Ape1 was shown (Fig 6) to stimulate excision of 8-oxoG and eC by increasing Ogg1 and Tdg turnover on damaged DNA . This can explain the increased repair capacity for 8-oxoG and eC of in cardiac muscle of pigs in groups fed a balanced diet (BDG), unbalanced diet (UDG), and pigs fed 9 moths unbalanced diet, followed by 3 months balanced diet - regression group (RG). Even from 3 months to 9 months after diiferent type of diet induction, in spite of the fact that glycosylases’ mRNA level did not differ significantly from that in control animals. eA glycosylase, being monofunctional, requires Ape1 activity, and therefore in the assay used also depends on its availability. Transcription of Ape1 is known to be induced by oxidative stress , and was shown to increase also in many types of human cancers. This suggests that treatment of pigs with a balanced diet (BDG), unbalanced diet (UDG), and pigs fed 9 moths unbalanced diet, followed by 3 months balanced diet - regression group (RG), induces long-term oxidative stress, which not only may increase DNA damage, but also induces repair processes. Acceleration of DNA repair, similarly to its inhibition, may, nonetheless, be a deleterious phenomenon. These results are consistent with the hypothesis that the adaptive and imbalanced increase in MPG and APE1 is a novel mechanism contributing to MSI, and, more generally, to genome instability, and in consequence may extend to other diseases with MSI, e.g., to cancer. Our results seem to confirm this hypothesis, suggesting that imbalanced DNA repair contributes to increased genomic instability and formation of preneoplastic lesions in cardiac muscle of pigs in groups fed a balanced diet (BDG), unbalanced diet (UDG), and pigs fed 9 moths unbalanced diet, followed by 3 months balanced diet - regression group (RG). Thus, the different type of diet can be -induced oxidative stress, as well as imbalance in repair processes in childhood may be a risk factor for developing colon cancer in adulthood.
Round 2
Reviewer 1 Report
Comments and Suggestions for Authors
The authors have not fully and correctly addressed all of my comments and the manuscript still has serious flows.
In particular:
- the regimen is not clearly presented: 3 groups control normal diet, high fat western diet and the western diet than regression to normal diet. This is not presented as clearly in the text.
- the figure 6 is still incomprehensible, normally in the study 3 groups and in this figure 6 genes. Only 6 bar are presented in the figure.
- many comments have been added in the results which may be in the discussion
- finally, the main scientific message of the manuscript is not clearly defined and presented.
Thus for all of these reasons, I maintain my decision to reject this manuscript.
Comments on the Quality of English LanguageN/A